# The Difficulty of Measuring the Roughness of Glossy Surfaces Using the Triangulation Principle

Juraj Ružbarský

Faculty of Manufacturing Technologies, Technical University of Košice, Štúrova 31, 080 01 Prešov, Slovakia; juraj.ruzbarsky@tuke.sk

**Abstract:** In the experiment, the roughness was measured on a machined surface with high gloss, which was also the main requirement for the test samples. For this reason, the samples made of stainless steel A304 and aluminum alloy AW 2017 were created by a progressive laser using material-cutting technology. This article explains a contact-free measurement method that uses the triangulation principle, which constitutes the basis on which the device used in the experiment, i.e., the laser profilometry, works. The surface roughness of the cut surfaces was examined on the manufactured samples through the selected roughness parameters of Ra and Rz. These parameters are commonly used in industry to quantify the roughness of a surface. The values measured in a contact-free manner were then compared with the reference values measured in a contact manner. Data from individual experimental measurements were graphed as dependencies based on which problem areas of measuring the roughness of glossy material surfaces with laser profilometry were described. Laser profilometry is a non-contact method for measuring the roughness of surfaces, and given the presented results of the experimental measurements and selected roughness parameters of the cut surface using a laser, we do not recommend using it for materials that have a glossy surface.

**Keywords:** profilometry; laser; glossy surface; roughness; stainless steel; aluminum alloy

## 1. Introduction

Increasing the service life and functional reliability of components depends on the high quality of their functional surfaces, i.e., dimensional and shape accuracy and surface roughness. With the increasing use of progressive construction materials (generally with impaired machinability), the development of cutting materials and machine tools requires, among other things, the development of measurement technology and evaluation methods for the quality of component surfaces [1].

Roughness parameters are only one of many factors affecting machining quality, which has a direct impact on productivity. However, higher roughness values are undesirable in terms of mechanical vibrations that can cause noise and dynamic loads. Subsequently, these vibrations can cause fatigue, malfunctions of the machine structure or its functional parts, energy losses, and reduced performance. Therefore, it is important to control and minimize roughness to maintain machine integrity and achieve optimal performance. By eliminating unwanted roughness, the machine operator can minimize the risk of these negative effects and increase the overall efficiency and effectiveness of the machining process [2,3].

The measurement and analysis of surface roughness in the machining process have several important aspects. Undesirable surface roughness can signal damage to functional parts of the machine tool, wear of the machining tool, workpiece, cutting head, or other machine parts. These processes can cause damage to the machine structure, its rigidity, bearing components, or other parts used in the machining process. Therefore, it is important to understand and control the surface roughness in the machining process to maintain the efficiency and durability of the machine tool and the quality of the machined parts [4,5].

Through the evaluation of roughness parameters, it becomes possible to predict the necessary sequence of operations to achieve the desired surface quality and to optimize the process of surface creation [6].

The value of surface roughness is often a critical factor for products that come into contact with each other [7]. It also has a significant impact on the durability, reliability, and operation of technical equipment [8,9].

The emerging surface roughness is not only a carrier of partial information but above all an image of the technology used to create it [10].

The clear trend of automating production processes also affects the automation of surface structure control. Nowadays, a common solution is to compare the measured surface with the sample surface and then evaluate it for technological modifications that should ensure the desired surface of the product. The ideal solution in fully automated production is to find out the resulting deviation of the surface from the defined standard and then adapt the process by changing the machining corrections to achieve the required surface structure quality [5].

When evaluating surface quality, not only roughness parameters are taken into account, but also dimensional uncertainty, geometric shape and position, and surface hardness of components. These surface characteristics are studied in the field of topography, which deals with the description of the surface of the studied object [11].

The 3D measurement and evaluation of surface topography (Figure 1) are new possibilities for how to express and present the surface structure. Due to the additional dimension and increasing amount of data, there is a possibility of more objective surface evaluation, possible prediction of functional properties of the surface, and their gradual modification during operation. Subsequent spatial analyses of the measured surface structure provide a graphical representation of the profile in various views in the form of a topographic map or a record of the coordinate's intensity. This form of measurement yields a large amount of information that an experienced observer can use [12].

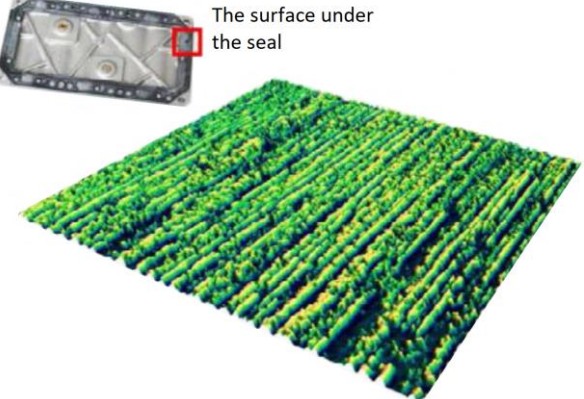

**Figure 1.** 3D structure of the gearbox cover.

3D surface topography uses spatial characteristics of the surface, which provide a large amount of information for a truly realistic representation of the surface under examination. Surface characteristics, e.g., the count, size, shape, and distribution of protrusions and profile recesses, not only increase the objectivity of surface evaluation but can also be used to predict the properties of functionally loaded surfaces and their changes during operation and solve several problems in tribology and production technology. Expanding the possibilities of the quantitative evaluation of geometric and dimensional changes in the surface profile at various stages of the components' production processes or upon their completion and even during their function is one of the main advantages of the spatial evaluation of the surface structure. Thus, there is a clear shift from the original contact measurement to a contact-free measurement of a range of profiles, providing fast, touch-free, and accurate measurement of the surface topography of common machine

components, where the advantage is to display the surface roughness over the component's entire functional area [13].

The measurement and processing of a greater amount of data in spatial surface evaluation are made possible by a new generation of measuring devices that simultaneously stimulate research in many technical and other areas. The real image of the surface obtained contains a large amount of information that can be interpreted by an experienced observer. A quantified description of the surface is provided by precisely defined parameters. It should be remembered that the parameters cannot describe the surface comprehensively. Each parameter brings information only about certain properties of the surface structure and requires correct interpretation [14].

When comparing the spatial measurement of the surface texture with the evaluation of a single cut (profile) of the surface, it can be concluded that 3D surface texture measurement is more objective and gives more accurate information about the state of the surface with a greater statistical significance of the characteristic evaluated. The evaluated spatial parameters of the texture are based on a significantly larger amount of measured data and have a much higher reliability. Thus, the obtained value of the parameter is much more reliable. Using the spatial evaluation of the surface structure also significantly reduces the dangerous neglect of the influence of one of the essential properties of the surface [15].

Most of the spatial surface parameters are, by analogy, dependent on the surface roughness parameters in 2D. The classic 2D roughness measurement is still prevalent due to the standardized and unambiguous surface roughness parameters and because it constitutes the foundation of the entire surface metrology. The parameters such as the spatial density of the protrusion are often estimated from 2D measurements. The resulting value is usually 20% higher than in reality, but it is enough to convey an idea of a functional area. Scanning the surface for spatial evaluation is mostly carried out on parallel profiles [16].

Although several instruments for measuring and evaluating the spatial characteristics of the surface are currently available, it is clear that the practical transition from evaluating a single profile to evaluating a surface is a long-term process. The use of a qualitatively new method of surface evaluation will require solving or specifying the number of problems associated not only with the actual measurement of the surface, but also with the processing of the results and, above all, with the effective use of the results obtained. The success of the enforcement of spatial surface quality control in metrological practice will undoubtedly be determined not only by technical but also by economic aspects [17].

The advantages of the spatial assessment of the surface structure show that these are the metrological methods of the future. Their wider practical use will undoubtedly be supported not only by increasing demands on the accuracy and quality of the existing production technologies but also by the development of new materials and technological methods, such as nanotechnology, etc. [18].

The use of spatial surface inspection offers a demonstrably significant improvement in the properties of functional surfaces, as well as progress in production processes, e.g., energy savings, etc. Therefore, it is in the interest of the technical development of production and inspection technologies to create all the prerequisites for the comprehensive use of spatial surface evaluation in regular production and metrological practice [19].

Experimental setups for optical topography methods have a similar basic structure, which consists of an optical source (for easier setup and calibration of the experimental setup, a light source is often used, that is, an optical source shining at a wavelength of 380–760 nm), an optical detector and a measured object. The optical source generates a light beam that is projected onto the surface of the measured object through the so-called optical probe, which can be point, linear, area, or intense, and subsequently reflects and refracts. An optical detector, which usually consists of a camera with a CCD or CMOS sensor, captures the reflected light beam and generates an image that is subsequently processed and analyzed. Different surface analysis methods vary in the type of optical probe used and how it interacts with the surface being measured [20].

In various laboratory or industrial measurement applications, measuring and assessing objects' spatial form by the contactless measurement of their height or thickness of a 3D profile is essential. Laser profilometry is designed to carry out these tasks quickly and efficiently [21].

Contact-free measurement methods replace contact measurements of surfaces sensitive to mechanical damage, such as soft materials, etc. Contact-free measurement avoids damaging the measured surface, and the inspected surface is monitored by a focused measuring head. The program-controlled adjustment of the measuring head is quick and easy [1].

3D profilometry using a laser is one of the most recently developed approaches to measure surface roughness, while pulse thermography is used to combine the effect of surface roughness with thermal radiometry [22].

Laser profilometry uses a well-known triangulation principle (Figure 2). The principle consists of the projection of a linear laser beam onto the measured object and the detection of its deformed image on the measured surface by a camera on the computer. The light source, the measured object, and the camera form a triangle whose parameters (distance of the camera and light source from the reference plane, their angle to the reference plane) are known. In this way, each scanned image can be used to create a real 3D profile [23].

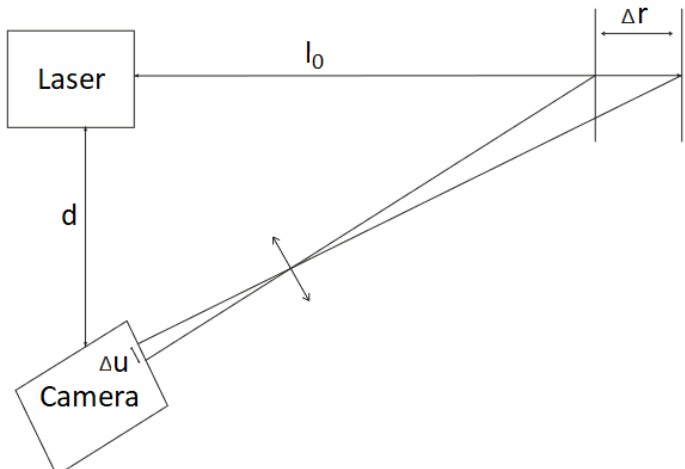

**Figure 2.** Scheme of the laser triangulation experimental setup.

The sought topographical deviation $\Delta r$ is the distance of each point of the measured surface whose coordinates correspond to the pixels of the camera chip matrix from the reference plane. This topographic distance is determined by the relation

$$\Delta r = \frac{\Delta u}{b + a\Delta u},\tag{1}$$

where:

$\Delta r$—sought topographic deviation,
$\Delta u$—the number of pixels on the matrix of the camera chip by which the display of the point is shifted during projection onto the reference plane and the measured point,
a, b—parameters of the mapping algorithm, which can be calculated from the known parameters of the experimental setup or obtained from the calibration of this setup.

The possibilities of profilometry do not end with the basic laser line–camera configuration. For example, a substantial increase in accuracy can be achieved in certain cases by using a projection grid instead of a line [11].

Laser profilometry is a non-contact optical method of measuring the profile of a scanned object surface, which then allows the creation of a 3D model of the object. This method utilizes a laser beam emitted by an optical source of electromagnetic radiation and

a sensing device in the form of a camera, the basis of which is a CCD or CMOS chip, to capture the profile of the object's surface [5].

Just like other methods of roughness measurement, laser profilometry has its pros and cons. One of the advantages of contact-free laser profilometry is that there is no wear on the measuring tool or damage to the surface of the measured object during measurement. It is capable of detecting cracks and micro-unevenness, which is a problem when using contact roughness meters that rely on tip geometry. Laser profilometry provides better repeatability of measurements and allows for measuring even highly curved surfaces. It can also evaluate multiple parameters of roughness and undulation and measure the entire surface area of the sample rather than just one line. Problems often associated with non-contact laser profilometry include difficulties in measuring glossy surfaces, problems in measuring multi-layered surfaces, the high cost of equipment acquisition, and the need for skilled operators [24,25].

After conducting a comprehensive study of the existing literature, the objectives of the paper were established. When choosing a specific topographical method, several criteria should be considered, and these criteria depend on the nature of the experiment. Some of the most important criteria to consider include the measurement speed, size of the measurement area, sensitivity of the method, accuracy of results, and financial difficulty. Part of the experiment was to assess whether laser profilometry was a suitable method for measuring the roughness of materials with a glossy surface or not. The test samples were made from stainless steel material A304 and aluminum alloy AW 2017 by cutting the respective material using laser technology. The main requirement for the samples was that their cleavage surface showed a very glossy surface when laser cleaved, and that was the reason why this cleaving procedure was used. The experiment of measuring the selected roughness parameters was carried out with a contact-free laser profilometer, and to verify the accuracy of the measured values and compare the measured values, the Mitutoyo Surftest SJ-400 contact roughness meter was used as well. At the end of the experiment, the experimental data from individual measurements were presented in graphs and analyzed. Another objective of the study was to compare two methods of measuring roughness: contact-free laser profilometry and the Surftest SJ-400 contact roughness meter. The experiment provided practical insights into the suitability of laser profilometry for measuring the roughness of glossy material surfaces based on the analysis of the sample surfaces.

## 2. Materials and Methods

The conducted experiments were aimed at monitoring the suitability of the contact-free laser profilometer for measuring the roughness parameters of glossy surfaces. The tested samples were made of stainless steel and aluminum alloy. The roughness on the cut surface was measured in a contact-free and contact way. Both methods were applied to the same cut areas measured. A Mitutoyo contact roughness meter (Surftest SJ-400, Tokyo, Japan) and a new commercial system on the market, a non-contact laser profilometer (KVANT, Bratislava, Slovakia), were used to measure and evaluate the samples.

In evaluating the surface roughness, the parameter Ra, the mean arithmetic deviation of the profile, and the parameter Rz, the tallest height of the profile unevenness, were monitored. In practice, the Ra and Rz parameters are those most monitored, capable of providing sufficient information about the quality of the measured surface [26]. The measured values of the sample materials were graphed as dependencies for both the Ra and Rz quality parameters of roughness.

### 2.1. Sample Preparation and Description of the Materials Tested

The sample production consisted of cutting the material with laser technology, which was implemented directly in the production plant of DRC s.r.o, Prešov, Slovakia where the TruLaser 3040 cutting machine from TRUMPF was used for cutting the material (Figure 3). The cut surfaces of the samples were very glossy once the samples were made.

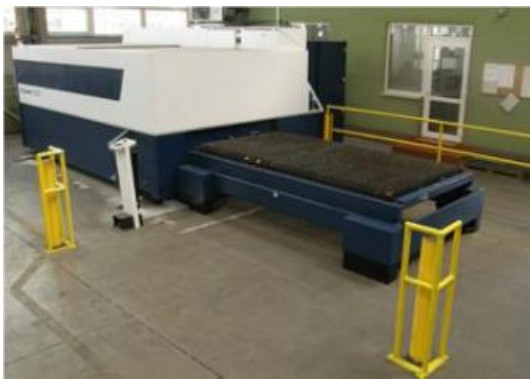

**Figure 3.** TruLaser 3040 material-cutting device from TRUMPF.

However, on glossy surfaces, there is an intense reflection of light that can cause some details on the surface not to be captured accurately. To verify the difficulty of measuring the roughness of glossy surfaces of materials with laser profilometry, samples made of stainless steel and aluminum alloy were used. These materials have different physical properties and may have different levels of glossiness and roughness. Using these samples allowed for comparing the measurement results with the expected results and identifying any issues in measuring the roughness of glossy surfaces.

Technical parameters of the cutting machine used to prepare the experimental samples are presented in Table 1.

**Table 1.** Technical parameters of the TruLaser 3040 cutting machine.

| Parameter | Value |
|---|---|
| Laser power | 3.200 W |
| Cutting speed | 2.1 m $\times$ min$^{-1}$ |
| Maximum cutting depth | 12 mm |
| Slice-gap width | 0.2 mm |
| Focal length | 4.5 mm |
| The diameter of the focused beam on the nozzle | 2.3 mm |
| The type and pressure of auxiliary gas used | Nitrogen (17 Bar) |

The analyzed materials were selected based on essential requirements such as high usability, availability of materials in the industry, as well as the various properties of the studied material (high gloss), which were taken into consideration during the experiment.

Material designation:

- stainless steel A304—material thickness 5 mm, material-cutting speed 2.1 m $\times$ min$^{-1}$
- aluminum alloy AW 2017—material thickness 5 mm, material-cutting speed 2.1 m $\times$ min$^{-1}$

The test samples were made by cutting the material using laser technology (Figure 4). The reason was that the material cut with laser technology manifests a highly glossy surface of the cut area.

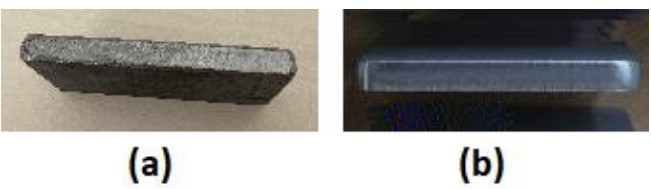

**Figure 4.** Experimental samples were used. (**a**) Stainless steel; (**b**) Aluminum alloy.

### 2.2. Measuring the Surface in a Contact-Free Way

Figure 5 shows the setup of the laser profilometer, which consists of basic and additional parts. The basic components included a sturdy aluminum support structure

comprising components for the vertical positioning of the measuring tube, a worktop mounted on stepper motors for movement along the *X* and *Y* axes, a laser beam source, a lens, and a camera equipped with a CMOS (complementary metal–oxide semiconductor) sensor and backlight. In our experiment, the light source was a laser blue diode StingRay with a wavelength of 445 nm. Additional parts of the laser profilometer setup included a computer with laser profilometry View control and evaluation software, as well as an image splitter.

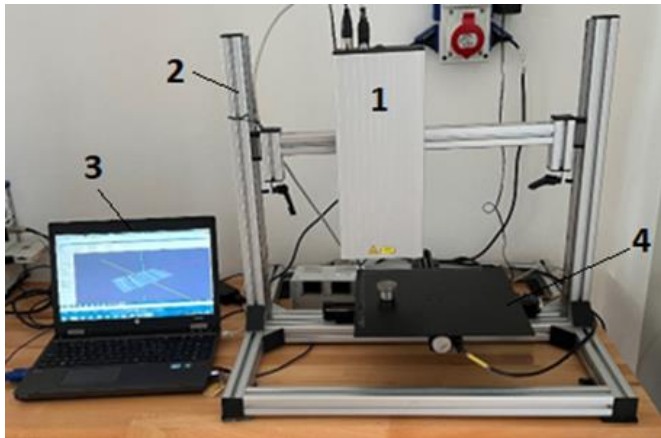

**Figure 5.** The assembly of the laser profilometer. (1) tube with laser and camera; (2) frame; (3) PC with operating and evaluation software; (4) worktop with step motors movable in *X* and *Y* axes.

When measured by the contact-free laser profilometry system, the experimental samples on the worktop were first placed in an anti-vibration mass (Figure 6). The first reason was that the machined surface of the sample was imperfectly fitted to the laser profilometry worktop and the sample on the worktop shifted during the movement of the stepper motors in the course of measurement. The second reason was to eliminate the influence of ambient vibrations, which can affect the quality of the measured data.

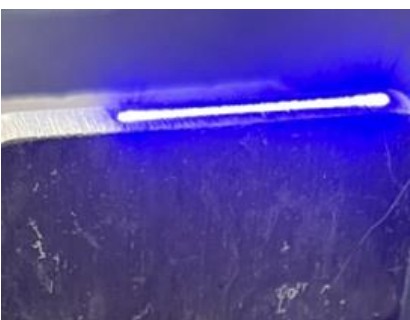

**Figure 6.** Measurement of the sample with the laser profilometer.

The input parameters for setting up the laser profilometry for measuring surface roughness are shown in Table 2.

**Table 2.** Input Parameters for Sample Measurement with the Laser Profilometer Setting.

| Axis *Y* Scan Distance | Axis *X* Scan Line |
|---|---|
| Scan lines count 45<br>Scan lines step 110 μm | 4000 μm |

The gain mode (image signal amplification in the camera) was set to 1. The shutter time (time of measurement) for the laser profilometer was set to 19.520 ms after previous

test measurements. This shutter time was found to produce the clearest image with the least amount of noise. By optimizing the shutter time, the laser profilometer was able to capture a clear and accurate representation of the surface of the sample. This allowed for precise measurements of roughness and other surface features to be obtained. This setup was consistent with the observations made by Srivastava et al. [27], Ružbarský [5], and Mitaľ et al. [26].

The experimental laser profilometry enables the measurement and evaluation of surface roughness parameters of samples by ISO 4287 standard (Rq, Rv, Rz, Ra, Rp). The measured roughness data can be exported in a CSV format, which is suitable for the further processing of experiments in the form of raw data. The surface roughness parameters evaluated using laser profilometry were Ra (arithmetic mean deviation of the surface profile), and Rz (the tallest protrusion in the profile's unevenness).

By following the recommended parameters, the laser profilometer was able to accurately measure the roughness of the sample surfaces. The parameters included in Table 2 may include the measurement range, the sampling frequency, and the scanning speed, among others, which can vary depending on the specific requirements of the measurement.

### 2.3. Measuring the Surface in a Contact Way

As mentioned in the introduction, to verify the correctness of the values measured with the contactless laser profilometer (Figure 7) and to compare them to those measured in a contact way, the measurement of samples was also carried out with the Mitutoyo Surftest SJ-400 roughness meter (Figure 8). Table 3 shows the individual parameters of the contact roughness meter.

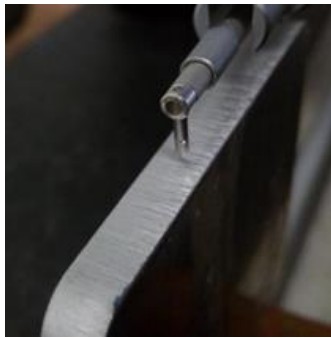

**Figure 7.** Measurement with the Surftest SJ-400 contact roughness meter.

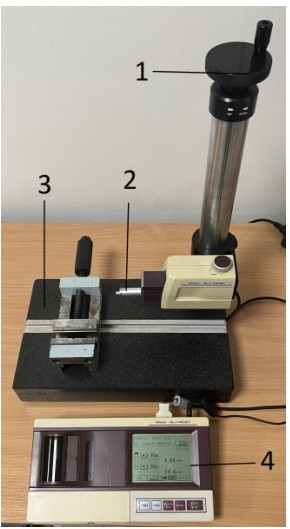

**Figure 8.** Surftest SJ-400 contact roughness meter. (1) Adjusting the standoff distance of the measuring tip; (2) measuring tip; (3) worktop; (4) touch display with the printer.

**Table 3.** Individual parameters of the Surftest SJ-400 roughness meter.

| Parameters | Value |
|---|---|
| Measuring speed (mm $\times$ s$^{-1}$) | 0.05; 0.1; 0.5; 1.0 |
| Speed of return (mm $\times$ s$^{-1}$) | 0.5; 1.0; 2.0 |
| Positioning | $\pm 1.5°$ (inclination); 10 mm (up/down) |
| Measurement range/resolution | P (primary); R (roughness); W (filtered waviness) |
| Evaluated parameters | 0.889 |
| Digital filter | 2CR; PC75; Gauss |
| Cutoff length (mm) | 0.08; 0.25; 1.8; 2.5; 8 |

The same part of the measured surface was selected for each sample for both measurement methods. The materials of the samples were defined according to the relevant standard as stainless steel A304 and aluminum alloy EN AW 2017.

*2.4. Experimental Samples Measurement*

Roughness measurement of the material surfaces cut with laser technology was carried out on samples of two types of material. The samples made from the two types of materials were of the same thickness and the cutting speed when splitting both materials was 2.1 m $\times$ min$^{-1}$. Each material was made from a single sample.

The measurement of the roughness parameters of Ra and Rz of the cut surface of the sample when measured with the laser profilometer and the Surftest SJ-400 was determined on each sample at three levels of measuring the cut surface, i.e., at the smooth, the medium and the coarse zone. In addition, the measured surface was transversally divided into parts A, B, and C (Figure 9). Individual measurement lines (1, 2, 3) were set to 1, 2.5, and 4 mm beginning in the smooth zone of the sample. The thickness of the sample material measured in both ways was the same.

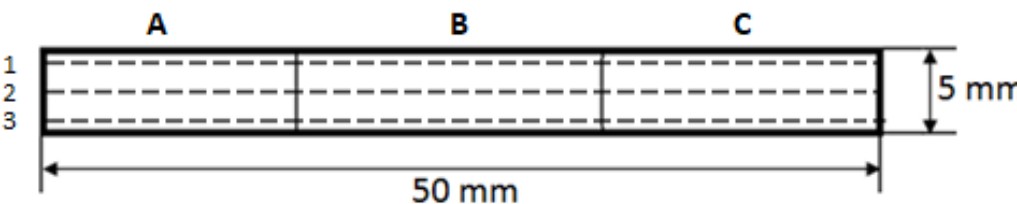

**Figure 9.** Samples with designated measurement lines and transverse zones.

The measurement itself was performed at several measurement lines, namely at a smooth zone of 1 mm, a medium zone of 2.5 mm, and a rough zone of 4 mm. Each of these measurement lines was divided into parts labeled A, B, and C, and three repetitions of roughness measurements were performed on each part.

**3. Results**

Due to the characteristics of glossy surfaces, measuring their roughness using contact-free laser profilometry can be challenging. The samples were made using laser-cutting technology. The roughness parameters values measured using the laser profilometry were then compared with the roughness parameters values measured with the Surftest SJ-400 contact roughness meter.

The graphical representation of the Ra parameter (Figure 10) illustrates the average surface roughness at different measurement lines (1, 2, 3) and parts of samples A, B, and C (length of the measuring section) for stainless steel and aluminum alloy, measured using contact-free and contact methods. These results were consistent with the conclusions reached by Jurko et al. [21], Foldyna et al. [28], and Azhari et al. [29].

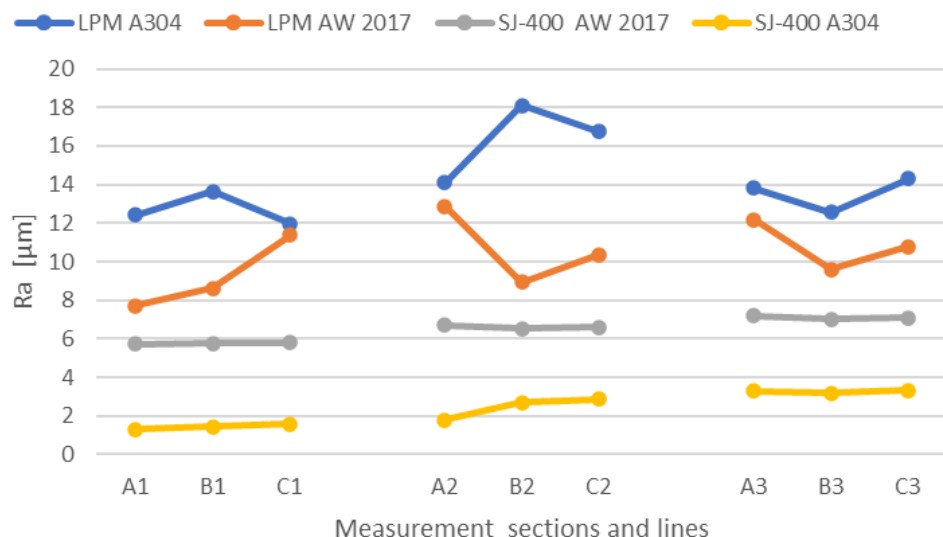

**Figure 10.** Graphed dependence of the average Ra values on the length of the measurement lines (1, 2, 3) and section (A, B, C) for stainless steel A304 and aluminum alloy AW 2017.

A comparison of the graphed dependencies involving the Ra parameter, which was measured in individual samples in a contact-free and contact manner, shows an obvious difference if we do not take into account the material used for the production of the samples (Figure 10). The measured Ra parameter values in the A304 material using the contact-free laser profilometer ranged from 12 to 18 μm, while the reference values measured with the contact roughness meter ranged from 1.6 to 2.5 μm. Similarly, the measured Ra parameter values in the AW 2017 material using the contact-free laser profilometer ranged from 8 to 13 μm, while the reference values ranged from 5.7 to 6.7 μm. This difference between the values measured with the contact-free method and the reference values are caused by the unwanted glare of the camera sensor, which leads to measurement error. However, this error can be partially eliminated by deleting the outlier values from the exported tables. These conclusions are supported by the findings of Stojanovic et al. [30] and Mital' et al. [31].

The 3D model shown in Figure 11 was generated by combining multiple profiles measured by the laser profilometer using the laser profilometry View software. In the image, the blue arrow indicates areas on the sample where the laser light reflected most strongly onto the CMOS camera. The software interpreted these areas as having the greatest surface roughness and reported them as sharp increases in the Ra parameter values.

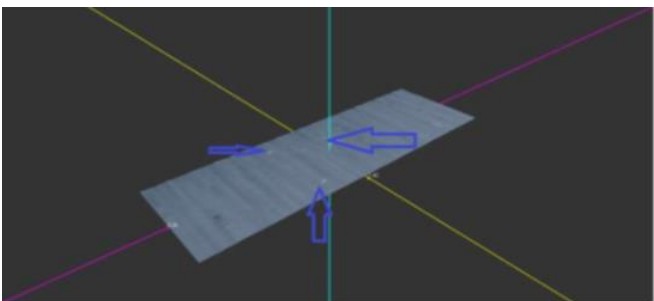

**Figure 11.** 3D model of the scanned sample made of AW 2017 (section A, 5 × 22 mm). *X*-axis—red line; *Y*-axis—yellow line; *Z*-axis—blue line.

There is also a visible difference in the Ra parameter in the materials of the samples used that were produced by laser technology of material cutting. In the contact method, the value of the roughness parameter Ra of the AW 2017 aluminum alloy material is in the order of 4 μm greater than that of the stainless steel material sample. In general, we can conclude that the stainless steel used is significantly harder than the aluminum alloy,

which means that the roughness we achieve on the surfaces cut by laser-cutting technology is better for harder materials.

In the contact-free method, the measured values of the roughness parameter Ra are higher for a sample made from stainless steel than from the aluminum alloy. This is due to the higher gloss of the cut surface of the stainless steel material than that of the aluminum alloy.

The graphed dependence of the Rz parameter (Figure 12) shows the average surface roughness at different measurement lines (1, 2, 3) and parts of samples A, B, and C (length of the measuring section) for stainless steel and aluminum alloy measured in contact-free and contact methods. These results are also confirmed by the conclusions reached by Akkurt [32], Hreha et al. [33], and Boud et al. [34].

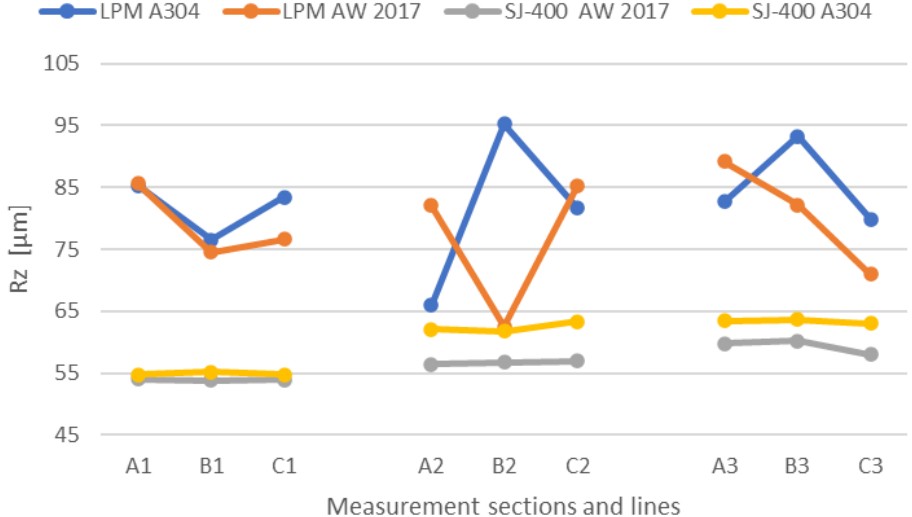

**Figure 12.** Graphed dependence of the average Rz values on the length of the measurement lines (1, 2, 3) and section (A, B, C) for stainless steel A304 and aluminum alloy AW 2017.

A comparison of the graphed dependencies involving the Rz parameter, which was measured on individual samples in a contact-free and contact manner, shows an obvious difference if we do not take into account the material used for the production of the samples (Figure 12). The measured Rz parameter values in the A304 material using the contact-free laser profilometer ranged from 65 to 95 μm, while the reference values measured with the contact roughness meter ranged from 55 to 67 μm. Similarly, the measured Rz parameter values in the AW 2017 material using the contact-free laser profilometer ranged from 64 to 89 μm, while the reference values ranged from 54 to 75 μm. We assume that the increase in Rz values measured by the laser profilometry view software compared to the contact roughness meter is due to the reflection of laser light from the surface of the sample to the CMOS camera. They also suggest that the Rz values measured by the contact roughness meter may be reduced because the diameter of the measuring contact tip could not traverse each surface depression. This is because the diameter of the measuring tip is larger than that of the surface depression. These results are consistent with the conclusions reached by Azhari et al. [35], Mitaľ et al. [31], and Ružbarský et al. [5].

There is also a visible difference in the Rz parameter in the materials of the samples used, produced by the laser-cutting technology. In the contact method, the value of the Rz roughness parameter of the AW 2017 aluminum alloy and that of the stainless steel is approximately the same. In general, we can state that the stainless steel used is significantly harder than the aluminum alloy, which means that the Rz parameter, obtained on the surfaces cut with the laser material-cutting technology, is better in the smooth and medium areas of softer materials.

So, in the contact-free method, the measured Rz values were similar in both materials, with only a slight difference observed in the medium area of the cut surface. This difference

may be due to the reflection of the laser light from the surface back to the CMOS camera, which is influenced by the material's glossiness. These findings are consistent with the results obtained by Mitaľ et al. [31] and Krenický [36].

## 4. Conclusions

Overall, the goal of the experiment was to determine whether contact-free laser profilometry is an accurate and reliable method for measuring the roughness of glossy surfaces. The results of the experiment can have important implications for industries that rely on precise measurements of surface roughness, such as aerospace, automotive, medical device manufacturing, and others. The obtained measurement results can be summarized as follows:

- The principle of the penetration of reflections into the measured data is based on the penetration of the reflected laser beam from the glossy surface of the sample into the CMOS camera, which causes an increase in the measured data compared to reality.
- Glossy surfaces can cause reflections of laser light, which can affect the accuracy of contact-free measurements of surface roughness parameters such as Rz. These reflections are captured by the CMOS camera and can be erroneously evaluated by the laser profilometry software as increased surface roughness. Therefore, it can be challenging to obtain accurate measurements of surface roughness parameters using a contact-free method for surfaces with high gloss. In such cases, contact methods may be more appropriate for obtaining accurate measurements (Figures 10 and 12).
- The stainless steel used is significantly harder than the aluminum alloy, which means that the Ra parameter obtained from laser-cut surfaces is better in harder materials (Figure 10).
- The 3D model created using the laser profilometry view software allows for a more detailed examination of the surface of the sample and can pinpoint areas where the laser light is reflecting the CMOS camera, leading to increased surface roughness values. The software evaluates these reflections as the greatest surface unevenness and records them as sharp increases in the Ra parameter (Figure 11).
- One way to partially limit this phenomenon is to coat the sample surface with a special transparent coating that absorbs some of the reflections, or the surface can be smudged or made matte. However, any such "finish" on the surface can distort the measured surface roughness results.
- We assume that another option to reduce the gloss of the material is to use a different color of laser light. However, it is necessary to compare individual colors of lasers as they affect the accuracy of surface roughness measurements.
- Previously used spatial (3D) surface assessment (x, y, z) can be supplemented with the fourth "dimension", variable (t), which can be time, temperature, pressure, or other physical parameters, the so-called 4D progressive analysis of surface changes. In this way, it is possible to evaluate not only the static properties of the examined surface but above all to monitor changes in surface texture, the course of action in the functional process, the impact of wear, deformation, cracking, erosion, and other structural changes.

The disadvantage of the SJ-400 contact roughness meter compared to the laser profilometry device is that it does not provide a complete picture of the roughness on the evaluated surface or the roughness in individual parts of the evaluated area. The disadvantage of the contact-free laser profilometer is that it is not able to reliably scan parameters from a glossy surface, from which laser light is reflected into the scanning camera, and subsequently, the laser profilometer software evaluates the reflection points as surface roughness problem areas. Laser profilometric measurement is a modern and reliable method of obtaining information about the surface of the measured object. This is thanks to the possibility of obtaining detailed data on the surface profile. One of the advantages of the laser profilometer is that it can obtain information about the largest depression and the largest protrusion of the surface. This allows for obtaining more detailed information

about the surface of the measured object, which is useful in various applications. This is made possible with the help of high-precision targeting of the laser beam and subsequent processing of the obtained data. Another advantage of laser profilometry is the ability to capture a series of surface profiles at different points, which allows it to generate a 3D image of the surface that allows a better view of the surface character of the object. This 3D image provides a more detailed view of the surface, including any problem areas caused by reflections of the laser beam.

A more detailed evaluation of the individual roughness parameters that were selected depending on the increasing feed rate could not be carried out precisely because of the very small number of samples, and thus the small volume of necessary data. As can be seen in Figures 10 and 12, this area deserves attention not only because of the changing roughness parameters Ra, and Rz, the changes in which are noticeable at first glance but also because of the changing composition of the surface roughness profile.

**Funding:** This research was funded by the Cultural and educational grant agency of the Ministry of Education, Science, Research and Sport of the Slovak Republic (KEGA), grant number 015TUKE-4/2022.

**Institutional Review Board Statement:** Not applicable.

**Informed Consent Statement:** Not applicable.

**Data Availability Statement:** Not applicable.

**Acknowledgments:** The author would like to thank the KEGA grant agency for supporting research work within project 015TUKE-4/2022.

**Conflicts of Interest:** The author declares no conflict of interest. The funders had no role in the design of the study; in the collection, analyses, or interpretation of data; in the writing of the manuscript; or in the decision to publish the results.

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
