# Peer review of "The Difficulty of Measuring the Roughness of Glossy Surfaces Using the Triangulation Principle"

_applsci, doi:10.3390/app13085155_

Round 1

Reviewer 1 Report

The article needs proofreading. English sometimes takes work to read and understand.

The images used in the article could be of better quality, especially images 1,2,8,9,10, and 12. Please improve the resolution of the images used and consider using vector graphics.

Lines 136-138 "As a detector, a CCD 136 camera or CMOS camera is most often used, and an optical probe is used as a light source, which is an optical structure falling on the surface of the measured object."

 This is a remarkably colored description for imaging a scene illuminated by a structured light illuminator. However, such a description may mislead the reader. Please rephrase that.

In the laser triangulation method, different geometries of the camera-illuminator system can be used. Different resolutions are possible depending on the geometry chosen, and a different effect of reflections on the image obtained is observed. There needs to be more information in the article about the geometry used. As this is strongly related to the research problem, please provide this information with a justification for the choice.

Is "LPM" the abbreviation of "Laser Profilometer Method"? If so, it is unnecessary to repeat it in phrases like "laser profilometer LPM" (lines 354, 393, 459, and more ).

Lines 164-168 "Laser profilometry (LPM) is a non-contact optical method of measuring the profile of a scanned object surface, which then allows the creation of a 3D model of the object. This method utilizes a laser beam emitted by an optical source of electromagnetic radiation and an image-reading device, typically based on a CCD or CMOS sensing chip, to capture the profile of the object's surface [5]."

 The author describes 3D imaging using the laser triangulation method. The "image-reading device" is called a camera or sensor. CCD or CMOS are types of sensors. To create a 3D image, moving objects relative to the vision system is necessary. Please add some information about the laser triangulation imaging system.

Is the LPM device used a commercial system, or did the authors develop it? Please add information about it. It is a crucial part of the system, but the device's parameters are merely mentioned in the Materials and Methods section.

In line 238 author describes the light source as a "laser red diode", please describe how you changed the wavelength in Figure 6 (presenting the measurement with the laser profilometer); the laser line is blue.

What is the method of calculating Ra and Rz in the profilometer? 

 Ra and Rz are calculated from a single profile or the 3D image.

 There need to be more parameters concerning the imaging, like resolutions, repeatability, etc. 

 Are digital filters applied in the profilometer like in the Surftest SJ-400 (table 2), or only RAW data is analyzed?

How are the parameters from table 1 (like the nozzle diameter of the cutting machine) relevant to the measurement of the roughness of glossy surfaces?

What is the effect of digital filtering used in the Surftest SJ-400? Please describe the applied algorithms.

Table 2 shows a parameter called "Measurement range/resolution (μm)" but no value is specified.

The cutting speed unit is invalid - "2.1. m.min-1." (lines 285, 222, 223 and table 1)

In section 2.4 author describes the measurement process. Is "Measurement level" related to the width of the profile line related to averaging? What is the reason for introducing this parameter? Please add some information about it.

Lines 301-338 are the description of the measurement process. Section "Results" should consist of gathered results - data. Please consider moving information regarding methodology to the Materials and Methods section.

Line 322 - how the exposure time was optimized? What were the objective function and constraints applied?

In table 3, what is the meaning of parameters in table 3? Please describe them.

About figures 10 and 12. What is the reason for creating a single line from all measurement points? Please reconsider the plot type because there is no connection between the value measured for points A4 and C2.5. Separation of measurement for "measurement level" and "traverse zone" could show additional dependencies. If not, why were those introduced?

What are the dimensions of the surface presented in figure 11? There are only 3 points with a glare which should be easy to filter.

Line 343, the plot in figure 10 is not a "graphical representation of the Ra parameter".

I suggest you read the article and the literature referenced to understand the issues of triangulation measurements better.

- Geometry and resolution in triangulation vision systems, Photonics applications in astronomy, communications, industry, and high energy physics experiments 2020.

The article requires a thorough correction and supplementation with information necessary for the discussed topic. 

Author Response

My answer to the reviewer is in the attached file.

Reviewer 2 Report

The article introduces measurement of roughness of AW 2017 and A304 by two kinds of instruments. The experiment results are obvious. But no further research has been carried out based on the results. For example, is it possible to reduce the reflection of light by different color lasers, and how about the test results in different laser powers, different test velocity, etc. So I think it is more like an experimental report than an academic paper.

Author Response

(The authors gave the same response as above.)

Reviewer 3 Report

Manuscript Number: applsci-2295652

Title: The difficulty of Measuring the Roughness of Glossy Surfaces Using the Triangulation Principle

Decision: Minor revision

Article Type: Article

The article is, in general, well written but there are some issues that article should consider to revise in order to improve its quality. Some comments were done in this way:

Ø  The abstract, according to the reviewer, is not a mini-paper but a quick tool to help readers decide whether they will read the rest of the paper. Please give the improvements that will attract the attention of the readers numerically (percentage rates) in the summary section.

Ø  Figure quality should be improved.. (Fig 1,2,4,5,6,8 and 11)

Ø  The stainless steel Ra value given in Figure 4.a is quite high. It cannot be measured with the Surftest SJ-400. The needle is attached. How did you measure the Ra value?

Ø  The Conclusions section should not repeat detailed information from the previous sections.

Ø  The literature is new but incomplete. Additions should be made from articles published in Applied Sciences journal.

After making the above corrections would recommend this article for publication in Applied Sciences.

Author Response

(The authors gave the same response as above.)

Round 2

Reviewer 1 Report

Rev.2

In line 276 author describes that the measurement is carried out in 200 steps with a step size of 0.11mm. The resulting length of the measurement line should be 22mm. In figure 9, the line length is 50mm, cut into three parts: 50mm/3=16.7mm. On the plot (fig. 10) "length of the measuring section" is 16mm based on 9 points (resolution is 2mm).
Please decide which statement is true.

Autor declares that the resolution in the Y axis for the profilometer is 0.11 mm. What is the resolution for the X and Z axes? Please introduce more information about the compared devices, like accuracy and repeatability.

What is the difference between sections A, B, and C. What is the difference between lines 1, 2, and 3? In lines 331-334, it is described that the sample consists of smooth, medium, and rough zones. Why were those introduced if in the next stage, all of the data is averaged and they are omitted in results and conclusions?

 In table 3 author mentions three digital filters embedded in the Surftest SJ-400: 2CR, PC75, and Gauss. What is the influence of those filters on the resulting measurement? Are there any filters like that implemented in the profilometer?

Please comment if the differences in the values shown on the plots could be related to the applied signal processing in both devices.

"Overall, the goal of the experiment was to determine whether contact-free laser profilometry is an accurate and reliable method for measuring the roughness of glossy surfaces." (lines 418-412)

So, is laser profilometry an accurate and reliable method for measuring the roughness of glossy surfaces? What is the conclusion?

The article lacks detailed information on the resolution and configuration of the triangulation measurement system. As a result, the author does not delve into the operation of the triangulation system or its construction. This significantly affects the implementation of measurements and is of key importance from the measurement resolution point of view. It isn't easy to evaluate the results obtained from a commercial device without knowing what measurement algorithm was used to obtain the results.

These issues are described in detail in scientific papers: "Geometry and resolution in triangulation vision systems. Photonics applications in astronomy, communications, industry, and high energy physics experiments. 2020"

The work requires a description of the configuration of the triangulation system and the algorithm for determining the height of individual points. This is very important, especially when determining the roughness parameters with the resolution of micrometers.

In addition, the inaccuracies described above remained in work.

Author Response

My answers are in the attached file.

Reviewer 2 Report

The paper shows the experiment results that it is difficult to Measure the Roughness of Glossy Surfaces Using the Triangulation Principle. The result has certain reference for some industries.

Author Response

My answers are in the attached file.

Round 3

Reviewer 1 Report

The author presents the results of measurements made with the use of a commercially available tiangulation profilometer. In the corrected version, it also clearly indicates the conclusion resulting from the measurements.